# Oral Manifestations in HIV-Positive Children: A Systematic Review

**DOI:** 10.3390/pathogens9020088

**Published:** 2020-01-31

**Authors:** Dorina Lauritano, Giulia Moreo, Luca Oberti, Alberta Lucchese, Dario Di Stasio, Massimo Conese, Francesco Carinci

**Affiliations:** 1Department of Medicine and Surgery, Centre of Neuroscience of Milan, University of Milano-Bicocca, 20126 Milan, Italy; moreo.giulia@gmail.com (G.M.); luca.oberti@outlook.it (L.O.); 2Multidisciplinary Department of Medical and Dental Specialties, University of Campania-Luigi Vanvitelli, 80138 Naples, Italy; alberta.lucchese@unicampania.it (A.L.); dario.distasio@unicampania.it (D.D.S.); 3Department of Medical and Surgical Sciences, University of Foggia, 71122 Foggia, Italy; massimo.conese@unifg.it; 4Department of Morphology, Surgery and Experimental Medicine, University of Ferrara, 44121 Ferrara, Italy; crc@unife.it

**Keywords:** HIV, AIDS, oral diseases, children, highly active antiretroviral therapy

## Abstract

**Background:** The number of pediatric patients affected by HIV still remains high, mainly in developing countries, where the main cause of infection is vertical transmission from the mother. Even today, a large number of these children do not have access to treatment, and, without proper care, they die in the first few years of life. **Objective:** The aim of our review was to assess the prevalence of oral hard and soft tissue lesions in HIV-positive pediatric patients by identifying the most common manifestations and the overall impact that they may have on the children’s quality of life. **Study design:** A systematic review of the articles in the English language in PubMed and Scopus was conducted in March 2019 in order to identify the main epidemiological and cross-sectional studies on the topic. **Results:** Oral diseases are still one of the most common manifestations in HIV-positive pediatric patients, and they often represent the first form in which immunosuppression shows itself. An analysis of the literature shows that candidiasis is the most common oral lesion found in HIV-positive children. A significant incidence of gingivitis and gingival disease is also evident, though not strictly correlated to HIV infection. However, thanks to the introduction of new antiretroviral therapies, the incidence of HIV-related oral lesions is decreasing. **Conclusions:** An HIV-positive children care program should also include dental protocols, as oral disease negatively influences the quality of life, affecting both functional and social aspects.

## 1. Introduction

HIV infection could be defined as the major epidemic of our century, with dramatic human, social and economic implications. It is a chronic infection that is first characterized by an asymptomatic phase that can stay unchanged for years and, subsequently, by the appearance of the first symptoms due to immunosuppression. In the end, it can lead to acquired immunodeficiency syndrome (AIDS).

The cause of the symptoms is to be found in the destructive effect of the HIV virus on T-helper lymphocytes, in which the virus completes its replication cycle.

The Joint United Nations Programme on HIV/AIDS (UNAIDS) of 2018 recorded, at the end of 2017, 36.9 million HIV-positive people, 1.8 million of whom were pediatric subjects up to 14 years of age [1].

In addition to systemic diseases in seropositive subjects, oral lesions, due to the dysregulation of oral microbiota and the consequent development of opportunistic infections, are also frequently found [2]. 

Since it was detected in children for the first time in 1982, HIV infection has continued to afflict thousands of pediatric patients in the developing world: In 2017, 180,000 children under 15 years of age became newly affected by HIV, and only 52% of them had access to treatment.

Bearing in mind that the immune system of pediatric patients is still immature, children infected with HIV are exposed to sever conditions, such as multi-systemic disease affectation and faster disease development [3]. Fortunately, the number of HIV-positive children is falling significantly: Since 2010, new HIV infections among children have decreased by 35%, from 270,000 in 2010 to 180,000 in 2017 [1].

Even in children, it is very common to find oral mucosal lesions, which are often among the first manifestations of HIV-infection in pediatric patients with important prognostic values [4]. A poor oral health can reduce life quality, leading to difficult psychosocial and nutritional conditions and complicating the treatment of the systemic diseases [5].

Few data on oral and dental lesions exist in HIV-infected pediatric populations, but it is well known that, in children, the disease usually progresses faster and the outcome is more serious than in adults, resulting in a high mortality rate due to serious opportunistic infections [6].

Fortunately, after the introduction of HAART (highly active antiretroviral therapy), both the mortality and morbidity of HIV patients has reduced, and the rate of HIV-related oral manifestations has decreased [7].

In children, in addition to oral mucosal lesions, there is an increase in the prevalence of dental diseases; the literature agrees that caries prevalence in HIV-infected children is higher than in healthy children [8].

### 1.1. Objectives 

The aim of this study was to revise and analyze the literature, selecting studies that were focused on oral lesions found in pediatric HIV-positive patients in order to highlight the most useful interventions to be implemented to prevent these infections and their complications.

### 1.2. Clinical Question (PICO)

P: A population of HIV-positive African, Indian and Brazilian children (0–18 years old).I: Investigation about oral and dental lesions in pediatric patients affected by HIV.C: Comparison between the oral manifestations in untreated HIV-positive children and those who underwent ART (antiretroviral therapy) or HAART.O: Overview on the most common oral manifestations in HIV-positive children in order to assess their impact on pediatric patients’ quality of life. Definition of appropriate care procedures, preventing the onset of these infections and avoiding their complications.

## 2. Materials and Methods

### 2.1. Protocol and Registration

To provide an appropriate search protocol, the methods and inclusion criteria of this systematic review were selected following the PRISMA (Preferred Reporting Items for Systematic review and Meta-Analyses) statement [9].

### 2.2. Eligibility Criteria

We examined the main studies on the prevalence of oral manifestations and oral lesions in HIV-positive pediatric patients. Only articles that met the following inclusion criteria were selected:They had to feature a sample of HIV-positive patients from the age of 0–18 years (with wide age range and without significant gender predominance).They had to feature a diagnosis of HIV performed according the World Health Organization (WHO) 1997’s criteria.They had to feature the absence of other systemic diseases.The articles had to have a cross-sectional study design that was written in English language.

Items that included population samples aged over 18 years old were not considered. Literature reviews and case-reports were excluded from our study.

### 2.3. Search, Study Selection and Data Collection Process

The following search items, combined with the Boolean term “AND,” were used to perform an electronic search in the PubMed, Cochrane Central Library, EMBASE and Scopus databases: oral lesions, oral manifestations, HIV infection (Mesh database terms) and HIV-positive children, and HIV-positive pediatric patients. The search began in November 2018 and ended on 31 March 2019.

Two reviewers independently assessed the eligibility of the articles. Data from the selected items were extracted by one of the authors (L.O.), and a second reviewer (A.L.) verified them. The disagreements between the reviewers were solved with discussions between the two authors in a joint session, and, whether or not an agreement was not reached, a decision was made by a third author (D.L.). Experts were employed in order to identify any unpublished data, and the reference sections of all the selected studies were analyzed to avoid the exclusion of relevant papers. 

Only the studies that met the following eligibility criteria were included: they had to be cross-sectional studies that were published in English language with the full text available, and they had to be performed on samples of pediatric patients that were selected and monitored after the year 2000.

Though the pediatric population is typically considered to be between 0 and 14 years of age, we decided to include older children up to 18 years of age. In most countries, such as Europe, Australia, Mexico, Canada and Russia, in fact, young people younger than 18 years are considered minor and, even if modes of infection are different in adolescents than in younger children or infants, the majority of the studies on this topic also included adolescents.

We selected clinical studies that aimed to report the characteristics of the pediatric population affected by HIV and information regarding their oral and/or dental health status.

In order to collect homogeneous data, we only included studies in which the evaluation of the oral health status of patients had been performed by an oral physician and the diagnosis of oral lesions had been carried out according to the WHO 1997’s criteria. The flow chart used for this systematic review is shown in Figure 1.

All the studies were analyzed to extract the following information: (a) The country where the study was conducted, (b) the number of HIV-positive children and the number of children in the control population, (c) the age of the children, (d) the route of transmission, (e) the number of patients undergoing ART or HAART, (f) the number of patients with any type of oral lesions, and (g) the prevalence percentage of each type of oral lesion. 

Regarding oral manifestations, we focused the review on the most significant ones that were reported in the updated classification of HIV-related oral diseases. Among these, those that are most often found in children are candidiasis (pseudomembranous, erythematous, and angular cheilitis), oral warts, herpes labialis, aphthous stomatitis, and necrotizing ulcerative gingivitis or periodontitis [10].

### 2.4. Quality Assessment 

The quality assessment of the selected studies was conducted by using the Newcastle–Ottawa scale (NOS) [11]. The highest and the lowest scores recorded were equal to 8 and 5, respectively (average score of 6.73). Almost all the samples that were selected by the included articles were truly or somewhat representative of the average in the target population, and all the items assessed the oral health of each subject by using validated measurement tools. Data were analyzed with appropriate and well described statistical tests (including confidence intervals and the probability level) by all the authors. Thus, the quality of the studies may be considered high. The evaluation of the quality of each included articles is shown in Table 1. 

## 3. Results

### 3.1. Study Selection and Characteristics

The first search of PubMed, Scopus, EMBASE, and Cochrane Central databases provided a total of 401 articles (236 in PubMed, 63 in Scopus, 39 in EMBASE and 63 in Cochrane Central), and, after deduplication, 360 remained.

Only articles published in the English language were selected: Literature reviews and the case-reports were excluded, and only cross-sectional studies were considered. In the end, only 76 articles were assessed and, after examining titles, abstracts and the full-texts of each study, 51 articles were excluded because they did not meet the eligibility criteria or because the full-text was not available. Finally, 15 articles were included in our review. The last excluded articles were eliminated because they did not meet the inclusion criteria. The quality assessment of the included studies was performed by using the Newcastle–Ottawa Scale (NOS).

The studies that were selected for the review analyzed large populations of HIV-positive children: Each research paper included, on average, 200 subjects and a total of 2762 HIV-positive children. The study with the smallest population sample was the one conducted by de Aguiar Ribeiro et al. (2013) (57 patients), while the study by Meless et al. (2014), with 420 patients, was the one with the largest sample.

In these studies, we could observe an equal sex ratio, with only a slight predominance of male subjects in the studies by Adebola et al. (2012), Olisovicz et al. (2015), Oyedeji et al. (2015), Ranganathan et al. (2010) and de Aguiar Ribeiro et al. (2013) (M: F = 1.2–1.5: 1) and of female subjects in the studies by Subramaniam and Kumar (2013, 2015) and Ponnam et al. (2012) (M: F = 1:1.2).

### 3.2. Results of Individual Studies

In relation to age, the subjects included in the studies could be divided into three groups: The first group of four studies included children aged between 2–6 months and 13–14 years, the second group of five studies included children ages 5–6 and 15–18 years, and a third group included children between 1–2 and 12–17 years. From the analysis of the characteristic manifestations of the patients of the three groups, no significant differences emerged.

Most of the examined children came from African countries (six studies) or from India (six studies), and only two studies were conducted in Brazil. This confirmed the fact that precisely these countries are those in which HIV is more widespread, especially among the pediatric population.

Only four studies investigated the main route of transmission of HIV infection: Vertical transmission represented the main mode (90% in the studies by Kumar et al. and Ranganathan et al., and 96.5% in the study by de Aguiar Ribeiro et al. Furthermore, Ranganathan et al. (2010) and de Aguiar Ribeiro et al. (2013) reported 3–5% cases of transmission for blood transfusion, and Adebola et al. (2012) reported that circumcision and prior surgery represented potential risk factors for the transmission of HIV (respectively, a risk of 19% and 1%).

In the studies included in the review, we could also observe that access to treatment for HIV-positive children is still difficult: Only in two studies did 100% of children undergo treatment (ART or HAART), this percentage varied between 80% and 90% in two others, and it varied between 50% and 70% in eight studies. The duration of treatment, as shown in Table 2, was reported only by few studies and was quite variable, from one month to nine years and with an average duration of four years. However, none of the analyzed studies used this value as a criterion in assessing the prevalence of oral lesions. 

Additionally, Oliscovicz et al. (2015) showed that 65.8% of the HIV-positive children that were included in the study had already developed AIDS.

The global prevalence of HIV-related oral lesions in the pediatric patients was quite variable from study to study, especially in relation to the type of therapy. From the analyzed studies, a prevalence value between 60% and 75% emerged. Although one study reported a prevalence of 99%, three studies reported values around 10–20%, and two studies reported values between 30% and 50%.

Unfortunately, most of the studies conducted on this topic did not present a control population that allowed for a comparison between healthy and HIV-positive children. Only the studies by Ponnam et al. (2012) and Baghirath et al. (2013) evaluated a non-seropositive control population in which the number of oral lesions was close to zero. These studies both compared a sample of healthy children with two samples of seropositive children, one undergoing HAART and the other being untreated, showing that patients on HAART had a smaller number of lesions compared to the untreated ones.

Among the HIV-related oral lesions included in the new classification, only a few showed a significant prevalence in pediatric seropositive patients.

In all the included studies, candidiasis represented the most common lesion. The only exceptions were the studies by Ponnam et al. (2012) and Oliscovicz et al. (2015), in which the most commonly reported lesion was that of gingivitis (22% and 15.3% vs 15% and 1.8% of candidiasis).

The studies by Ranganathan et al. (2010), Adebola et al. (2012), and Nabbanja et al. (2013) reported a value of candidiasis prevalence between 50% and 80%; Sales-Peres et al. (2012), Ponnam et al. (2012), de Aguiar Ribeiro et al. (2013), Meless et al. (2014), and Oliscovicz et al. (2015) reported a prevalence of 4–5% (range 1.8–11%); and Oyedeji et al. (2015), Kumar et al. (2013) and Rwenyony et al. (2011) reported values between 17% and 28%.

In the study by Baghirath et al. (2013), 5–12 years old children undergoing HAART were compared with untreated patients of the same age, and it was shown that the prevalence of oral candidiasis is twice as high in untreated children (34% vs 16%). Divakar et al. demonstrated that the percentage of oral candidiasis in HIV patients receiving ART for more than three years was equal to 2.8%, compared to those of untreated HIV patients, which was around 38.2%. 

Pseudomembranous candidiasis and angular cheilitis are the most common forms of candidiasis, while the erythematous form is less common—though it was reported in almost all the studies. 

Ranganathan et al. (2010) reported that about 50% of patients suffered from pseudomembranous candidiasis, 20.3% suffered from angular cheilitis, and only 16.5% suffered from erythematous candidiasis; the studies conducted by Subramanian also reported similar values, although, in these studies, the prevalence of angular cheilitis was much higher (96.6% and 81.9%).

For Rwenyonyi et al. (2011), 16% of seropositive children suffered from pseudomembranous candidiasis, and 5–7% suffered from erythematous candidiasis and angular cheilitis. Adebola et al. (2012) reported that the prevalence of pseudomembranous candidiasis was 26.7%, the prevalence of erythematous form was 8.6% and the prevalence of angular cheilitis was 43.8%.

Finally, Nabbanja et al. (2013) reported prevalence values of 50.5% for pseudomembranous candidiasis and 10.3% for erythematous candidiasis.

Though oral candidiasis is the most frequently found HIV-related oral manifestation, other oral lesions are also quite common. The analyzed studies showed that the second most common oral lesion in HIV-positive children is gingivitis. In both the 1993 European Clearinghouse classification and the new 2009 classification, simple gingivitis was not considered to be an HIV-related oral manifestation, unlike necrotizing ulcerative gingivitis (NUG) and the necrotizing ulcerative periodontitis (NUP), which instead represented characteristic HIV lesions.

The significant prevalence of gingival disease in HIV-positive children has been hypothesized to be related to infection, but a direct correlation has not yet been demonstrated. The main reason for the difficulty of establishing the exact role of HIV in the genesis of gingivitis in infected children is the lack of studies on the subject or the lack, in the studies on this topic, of an adequate control group that allows for a comparison of the incidence of this disease in infected and healthy children.

Gingivitis, as reported by the WHO, is in fact a very frequent disease in children, especially in the poorest areas of the world where it is mainly caused to poor oral hygiene and the presence of various types of infections that weaken the immune system.

The studies included in our review showed that gingivitis is the second most common manifestation in HIV-positive children: Many studies reported percentages of the prevalence to be 10–20% (range 8.3–22%). The studies by Meless et al. (2014), Nabbanja et al. (2013) and Oyedeji et al. (2015) reported lower values (2–4%), while the studies of Ranganathan et al. (2010) and de Aguiar Ribeiro et al. (2013) reported values of 37.9% and 57.9%, respectively.

A rarer condition is represented by periodontitis, which was found in only five studies, with values always below 10% (range of 0.3–7.7%).

However, it is important to specify that in the majority of the studies examined, no distinction was made about the severity of gingival or periodontal disease. For this reason, with the term “gingivitis,” many studies referred to all the diseases that involve the gums, including ulcerative necrotizing gingivitis.

Only a limited number of studies indicated NUG and NUP as specific manifestations, reporting a significantly lower prevalence (NUG = 0–2% and NUP = 0–1%) compared to that of simple gingivitis.

Typical HIV-related oral manifestations, such as oral hairy leucoplakia (OHL), which is very common in adult seropositive patients, seemed to be not so frequently found in HIV-positive children. OHL was found by five authors, and the percentage of prevalence was always below 6% (range 0.7–6%). Furthermore, in patients undergoing HAART, the prevalence was significantly lower.

Another disease reported by the studies included in the review was linear gingival erythema (LGE), which was described by seven of the included authors. Prevalence values were similar in all these studies, and they were between 5% and 9%. This condition was also rarely found in patients undergoing HAART. LGE has always been considered an HIV-related lesion: It was originally believed that LGE presented a direct association with HIV, so much so that it was called HIV-associated gingivitis.

The new classification no longer includes LGE in HIV-related oral manifestations, reporting it as a form of *Candida* infection or classifying it with gingivitis.

Many of the analyzed studies, in fact, did not report precise data about the LGE prevalence, including this manifestation in the group of gum disorders.

Herpetic lesions were also found in some children, showing a prevalence between 0.5% and 8.6%, with an average value of around 2%.

Other frequently found manifestations are ulcers (recurrent stomatitis, canker sores, non-specific ulcers), which were reported in almost all studies: The global prevalence is lower than 15% (range of 1–14.3%). Only the studies by Subramaniam and Kumar (2013 and 2015, respectively), in which the prevalence of ulcers settled around 30% (33.3% and 30.3%, respectively), represented an exception.

Kaposi’s sarcoma is rarer in children, with incidence rates between 0.4% and 3.3%, while a more common lesion in children is the depapillation of the tongue: Subramaniam and Kumar (2013 and 2015) reported rates between 20% and 22%, Ranganathan et al. (2010) reported a value of 5.7%, and Kumar et al. (2013) reported a value of 9.2%.

Parotid gland enlargement, despite being reported only by a few authors, seems to be very frequent, affecting a very high percentage of HIV-positive children.

Finally, hyperpigmentation and oral warts are manifestations that have often been reported in literature and were reported in the analyzed studies. Their incidence appeared to be significantly greater in subjects undergoing HAART.

Values between 6.1% and 17% were recorded by the selected articles. Baghirath et al. (2013) showed that no pigmentations were found in untreated patients, while 10% of HAART patients had oral hyperpigmentation. Oral warts, on the other hand, are not so common: Only three studies reported them between HIV-related oral manifestations, and their prevalence was less than 5%.

For these reasons, such lesions could represent possible side effects of therapy. Major details about the data extracted from the included items are reported in Table 2.

Some studies have also evaluated the dental health of HIV-positive children, showing that the incidence of carious lesions is high. Nabbanja et al. (2013) recorded a prevalence of caries of 54.1%, Ponnam et al. (2012) recorded a prevalence of 29%, and Oyedeji et al. (2015) recorded a prevalence of 12.1%. The latter also reported that 44.8% of children had enamel hypoplasia. The Decayed Missing Filled Teeth indices of permanent and deciduous teeth(DMFT and dmft respectiveley) were also found to be higher than those of healthy children. Among the five studies that evaluated the DMFT index, an average value of 2.7 was reported, while the dmft index had an average value of 6.1. The values recorded by these studies are illustrated in Table 3.

This could prove that dental health is also compromised in HIV-positive children; however, it should be noted that none of the reported studies carried out a comparison with a control population, instead limited themselves to reporting a comparison between the data that were collected and the average prevalence values of caries and DMFT indexes reported by the WHO. In this way, they did not take into account the considerable differences related to the style of life, the environment, and the socio-economic conditions of children with HIV who come from the poorest areas of the world.

## 4. Discussion

HIV infections in children are still a serious problem in developing countries: from the data that were collected in the selected studies, it emerged that, while sexual contact represents the principal way of transmission in adults, the first cause of infection in children is transmission from the mother. The vertical transmission may occur transplacentally (during pregnancy), during delivery (as the infant passes through the birth canal), or postnatally (during breastfeeding) [21].

The immature immune system of children predisposes them to a rapid and fulminant disease process: About 50% of the HIV-positive children were found to be manifest clinical symptoms in the first year of life, and many of them showed initial signs of AIDS before one year of age [12].

HIV-related oral manifestations include a large group of lesions, like bacterial, viral, or fungal infections, as well as idiopathic lesions, salivary gland diseases and neoplastic diseases [23].

The prevalence of HIV-related oral manifestations is estimated to be in a range between 30% and 80%: This value increases when the CD4+ T-cell count is low (<200 cell/mm3) and the viral load is high (>20,000 copies/mL) [24]. Furthermore, the prevalence of oral lesions is greater in developing countries because of late diagnosis and difficult access to treatment [25,26]. 

In regard to oral lesions, they still represent one of the most common manifestations in HIV-positive pediatric patients, a fact that emerged from the articles included in this systematic review.

The EC-Clearinghouse on oral problems related to HIV infection produced a classification of HIV-related oral manifestations. Seven cardinal lesions are strongly associated with HIV: oral candidiasis, oral hairy leukoplakia, Kaposi’s sarcoma, linear gingival erythema, necrotizing ulcerative gingivitis, necrotizing ulcerative periodontitis, and non-Hodgkin’s lymphoma. These manifestations involve more than 50% of people with HIV infection and about 80% of those with a diagnosis of AIDS [27]. Similar data have also been found in children, with a very high incidence of oral lesions.

Recently, in 2009, a new classification was produced in which LGE was excluded and more emphasis was given to other manifestations such as herpes labialis, recurrent aphthous stomatitis and oral warts [10]. The main reason for LGE exclusion is that recent studies have revealed the existence of a relationship between LGE and the colonization of *Candida* species in the subgingival plaque. For this reason, it is possible to conclude that LGE is another variant type from candidiasis and not a distinct disease [28,29].

It is known that most diseases affecting HIV-positive children rarely occur in a healthy child. Gingivitis, ulcerative necrotizing periodontitis, recurrent aphthous stomatitis, angular cheilitis and oral hairy leucoplakia are lesions that are characterized by a very low incidence in healthy children, as demonstrated by several studies.

This is one of the main reasons that explains why in the majority of epidemiological and prevalence studies on oral lesions in HIV-positive children, there is no control group of healthy patients. Among the studies included in our review, only two presented a control population, and even in the epidemiological studies and literature reviews that were conducted more than 10 years, ago it was difficult to find studies that include a control group.

The workshop report by Arrive et al. explained that the reason of the absence of a control group is to be found in the fact that the oral manifestations of HIV are considered rare in healthy children or in any case with such a low prevalence that it does not make necessary to compare them with HIV-positive children [30].

In fact, the relationship between these typical lesions and the seropositive condition has been taken for granted. However, there are some typical manifestations of HIV that are easily found in healthy children as occasional findings. The most characteristic example is that of candidiasis which is typically found in many healthy children under six years of age.

This shows how it is necessary to carry out studies that include a control group that is characterized by children of the same ethnicity, of the same socioeconomic status, and with a similar lifestyle in order to correctly assess which oral lesions are HIV-related in a meaningful way.

The fact that these studies have been conducted in developing countries, the only ones in which pediatric HIV is still widespread, makes it difficult to collect accurate data and gain access to larger population samples, especially with regard to healthy children, who are hardly brought to the attention of hospitals.

Moreover, the studies included in the review often reported discordant values: This is due to the limited number of studies on the subject, the small size of patient samples, the differences in geographical areas from which the infected children come from (mainly Africa and Asia), and the differences in the methodology used for the collection and analysis of data (in fact, most of the studies do not report in a detailed and precise way the procedures that were used for the diagnosis of oral lesions and for the execution of statistical analysis).

The majority of studies conducted around the world agree that candidiasis (OC) is the most common HIV-related oral lesion, both in patients undergoing HAART and in untreated ones [31]. Two forms of OC are mainly observed: pseudomembranous candidiasis (PC) and erythematous candidiasis (EC). A third form, angular cheilitis, is also common in HIV-positive patients. These data are in agreement with what emerged from this review.

The types of oral lesions observed in children are similar to those found in adults except for the lower frequency of oral hairy leucoplakia, Kaposi’s sarcoma, and non-Hodgkin’s lymphoma, and for the higher prevalence of chronic parotid enlargement [19].

A controversial case is that of gingivitis, which is not considered an HIV-related lesion but appears to have a significant incidence in HIV-positive patients.

The reasons are to be found in the fact that (1) many authors include lesions such as LGE and NUG, typical of HIV infections, in the term “gingivitis;” (2) the presence of concomitant oral lesions due to the virus, which makes it difficult the oral hygiene procedures because of pain; and (3) the decrease in immune defenses makes the patient more susceptible to developing infections.

Nowadays, since the number of studies comparing a population of infected patients and a population of healthy patients with similar characteristics is limited, it is not possible to establish whether there is a significant relationship between HIV and gingivitis in children, an issue also caused by the fact that gingivitis is a common disease even in healthy children.

Moreover, some authors have argued that the weakening of the immune system by HIV would not be correlated to the onset of gingivitis, as it would only be correlated to the severity of the gingival lesions [32].

In children, on the other hand, a greater prevalence of carious lesions seems to be evident, with a dental health that is generally poorer than the one of healthy children [33]. The process of HIV infection include several factors that can cause dental caries: the prolonged use of sugary products and carbohydrates, changes in salivary flow and salivary glands (caused by use of drugs), immunosuppression, repeated episodes of hospitalization, poor oral hygiene, and poor competence in terms of oral health promotion [13].

The main problem remains the absence of articles that analyze the oral and dental status of HIV-infected children by comparing it with that of uninfected children. Though many studies have reported a significant incidence of caries in HIV-positive children, mainly for primary dentition, nowadays, the relationship between HIV infection and dental caries is still not clear [30]. It is not possible to establish whether the prevalence of caries or the presence of a high DMFT index may be related to HIV infection or if it does not differ significantly from the values recorded in a population of healthy children in the same countries with the same habits and the same socioeconomic status. In fact, the lack of control groups in the majority of the studies prevented a comparison on this topic, thus making it impossible to establish the existence of a correlation between dental status and HIV infection.

Oral and dental diseases also negatively affect the quality of life of these children: Most seropositive children suffer from an impact of oral problems on their daily activities, reporting many difficulties in eating and other functional limitations [34]. Furthermore, the impact that these problems have on emotional and social well-being, which is inevitably compromised, must be considered [35].

Therefore, the differences between HIV-positive children and adults make it necessary for oral physicians to use a diversified approach [36].

The main factors that influence the appearance of these lesions are the use of HAART, alterations of the CD4+ T-cell count, and high viral loads. Studies on children have also reported other factors, including poor oral hygiene and low socioeconomic status [5].

Nabbanja et al. (2013) observed that the incidence of oral manifestations in pediatric subjects with poor oral hygiene is significantly higher, while Baghirath et al. (2013) reported that children living in rural areas and those with a lower socio-economic status have more lesions than those living in urban areas or belonging to richer social classes [17,18].

The introduction of HAART, which is characterized by the association of proteases inhibitors and nucleosides reverse transcriptase or non-nucleoside reverse transcriptase inhibitors, has been one of the main factors in achieving a decrease of the prevalence of HIV-related oral manifestations in pediatric patients (12). Nevertheless, their incidence still remains very high [37].

In a prospective study by Ortega et al. (2009), a cohort of patients was followed from 1989 to 2006, and an important decrease was observed in the incidence of HIV-related oral manifestations right after the introduction of protease inhibitors (HAART-PIs) and when the first non-nucleoside reverse transcriptase inhibitors (HAART-NNRTIs) were diffused [38].

In 2017, 80% of pregnant women living with HIV had access to antiretroviral medicines to prevent the transmission of HIV to their babies, up from 47% in 2010 (1).

This is one of the reasons why life expectancy, which in the past was four years, has today increased to 20 years [16].

Most of the studies included in the review demonstrated that the prevalence of oral lesions tended to be greater in untreated patients [17,18].

Specifically, Rwenyonyi et al. (2011) and Adebola et al. (2012) observed that this difference was statistically significant only with regard to the incidence of oral candidiasis and gingivitis (14,15), while for Ponnam et al. (2012) and Subramaniam and Kumar (2013 and 2015), the difference in the prevalence of ulcers was also significant [4,16,21].

Despite the main goal of therapy being to get a decrease of viral load and a growth of CD4+ T-cell count, HAART also seems to alter oral innate immunity and the determination, in the long term, of a reduction of salivary flow rates [39].

This inevitably leads to a series of side effects, like hyperpigmentation, xerostomia and salivary gland hypertrophy, that have seen an increase in their incidence after the introduction of HAART. The cause for these effects is to be found in the effects of antiretroviral drugs on melanocyte-stimulating hormone (MSH) and on salivary flow rate [7].

Regarding this, the studies by Subramaniam and Kumar (2013, 2015) and Ponnam et al. (2012) showed that the prevalence of hyperpigmentation in children undergoing HAART was significantly higher [4,16,21].

Finally, another correlation that has been widely demonstrated in the literature and observed in the analyzed studies is that which exists between the CD4 + T-cell count and the presence of oral lesions [40].

During therapy, an increase in lymphocytes count leads to a decrease in the number of oral lesions: This is not statistically significant for every kind of oral lesions, as it is only significant in relation to oral candidiasis, as reported by Ponnam et al. (2012), Subramaniam and Kumar (2015), and Baghirath et al. (2013) [16,17,21].

Systemic and oral lesions in HIV infection reflect the immune status of the patients: In particular, the presence of oral lesions may be an early diagnostic indicator of immunodeficiency and could be used as predictor of HIV infection progression [7].

Furthermore, the resolution or the disappearance of oral lesions in HIV-positive patients can serve as indexes of treatment success.

## 5. Conclusions

Nowadays, defeating HIV infection in children represents one of the most difficult challenges in developing countries, where the virus is widespread. In addition to drastically reducing the life span of affected children and significantly affecting their quality of life, HIV also leads to the development of various systemic and oral lesions.

Specifically, oral manifestations such as candidiasis, oral warts, herpes labialis, and aphthous stomatitis, which represent some of the first signs of HIV immunodeficiency, could become, in these cases, very important diagnostic, and especially prognostic, indices; this is also due to the correlation existing between oral lesions, viral load, and CD4+ T-cell count.

From this review, it emerged that oral candidiasis is the lesion that is most often found, together with gingival lesions, mainly due to poor oral hygiene and therefore not strictly related to the infection itself. However, the use of highly active antiretroviral therapy has reduced the incidence of these lesions.

In HIV-positive children, it is essential to intercept the disease as soon as possible in order to start treatment quickly. The early recognition of these lesions and the implementation of adequate treatment could allow for the decrease of the high rate of child mortality caused by HIV/AIDS.

Moreover, it is essential to intervene on these children to solve oral and systemic lesions, guaranteeing a better quality of life. In fact, oral lesions are associated with discomfort during oral function, particularly during swallowing and tooth brushing, which may increase the tendency to maintain the poor clinical health of these infected children. To do this, it is necessary to investigate which the lesions are actually significantly correlated to HIV infection through studies on larger and heterogeneous samples that are compared with control populations of similar healthy patients.

Given the high prevalence of HIV infection, dentists should be aware of the latest developments regarding prevention and treatment of diseases, as well as the promotion and maintenance of the oral health of individuals with HIV/AIDS. For this reason, oral and dental care should also be integrated into pediatric HIV care programs in order to ensure regular screening for oral lesions and appropriate early management.

As it has been proven that oral candidiasis is the most common lesion in HIV-positive children, it would be useful to investigate the role that HAART has on the evolution of other lesions, to assess whether there is a link between HIV and gingivitis, and to investigate if the prevalence of oral lesions is concomitant with possible mediators (use of steroids or other drugs, specific systemic disease, etc.).

To do this, we need more accurate studies that include large population samples and control groups and that analyze and compare the different characteristics of the selected populations by investigating the existence of possible correlations with the onset of oral lesions.

## Figures and Tables

**Figure 1 pathogens-09-00088-f001:**
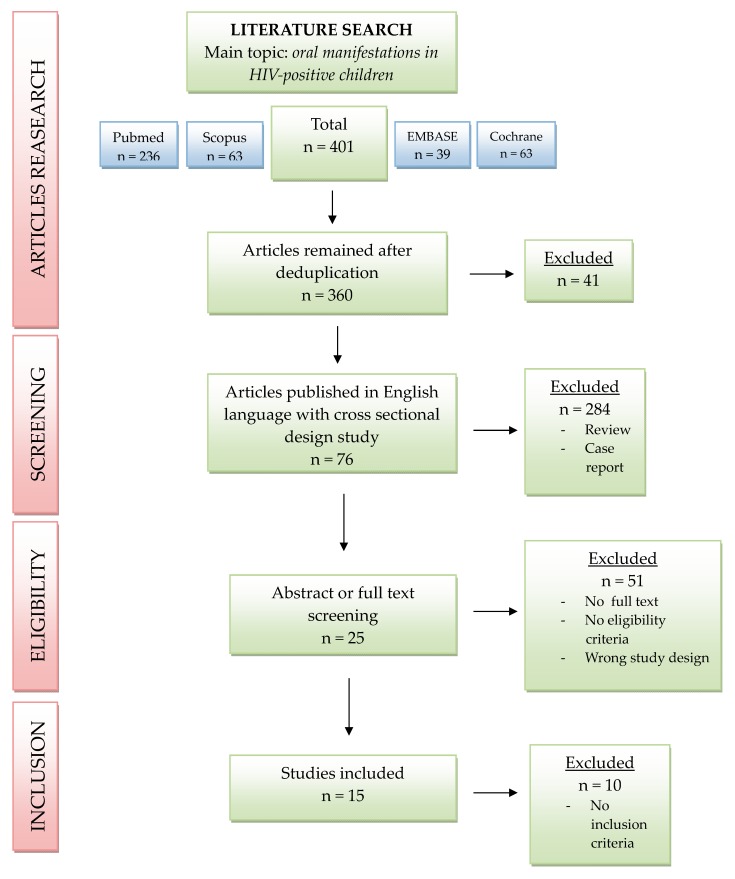
Flow chart showing the selection of studies for the review.

**Table 1 pathogens-09-00088-t001:** Quality assessment of included studies.

Studies	Representativeness of the Sample	Sample Size	Non-Respondents	Ascertainment of the Exposure	Comparability	Outcome	Total
Ranganathan, 2010	+	-	-	+	+-	++	5
Sales-Peres, 2010	+	+	+	+	+-	++	7
Rwenyonyi, 2011	+	+	+	+	+-	++	7
Adebola, 2012	+	+	-	+	+-	++	6
Ponnam, 2012	+	+	+	+	++	++	8
Baghirath, 2013	+	+	-	+	++	++	7
Kumar, 2013	+	+	-	+	++	+-	6
Nabbanja, 2013	+	+	-	+	++	+-	6
Ribeiro, 2013	+	-	+	+	++	++	7
Subramaniam, 2013	+	+	+	+	+-	++	7
Meless, 2014	+	+	+	+	+-	++	7
Oliscovicz, 2015	+	+	+	+	+-	++	7
Oyedeji, 2015	+	+	-	+	+-	++	6
Subramaniam, 2015	+	+	+	+	+-	++	7
Divakar, 2015	+	+	+	+	++	++	8

+ = star assigned; - = star not assigned.

**Table 2 pathogens-09-00088-t002:** Characteristics of the lesions that were detected in HIV-positive children.

Author, Year	Country	*n*	*ctr*	Age	ART (%)	Time of ART Use	Any OL (%)	OC (%)	G (%)	P (%)	LGE (%)	ULC (%)	HSV (%)	OHL (%)	PIG (%)
Ranganathan, 2010 [12]	India	212	0	1–14 yr	NR	NR	62.3	56.1 *(PC 50, EC 16.5, AC 20.3, HC 1.4)*	37.9	NR	NR	4.2	NR	1.4	NR
Sales-Peres, 2010 [13]	Mozambique	90	0	1.7–16 yr	82	NR	13.3	AC 4	NR	NR	NR	NR	NR	NR	NR
Rwenyonyi, 2011 [14]	Uganda	237	0	1–12 yr	49.8	NR	73	28.3 *(PC 16, EC 4.6, AC 6.8)*	10.8	NR	NR	1.2	1.7	NR	6.1
Adebola, 2012 [15]	Nigeria	105	0	2–156 mth	61.9	1–12 mth	61.9	79.1 *(PC 26.7, EC 8.6, AC 43.8)*	21.9	4.8	NR	14.3	8.6	NR	NR
Ponnam, 2012 [16]	India	190	95	5–15 yr	50	NR	ART: 32 noART: 52	PC+EC 4, AC 11	22	4	NR	9	3	NR	17
Baghirath, 2013 [17]	India	100	50	5–12 yr	50	12% <3 yr 21% >3 yr	ART:43	ART: PC+EC 14, AC 2 noART: PC+EC 32, AC 2	NR	NR	ART: 0 noART: 4	ART: 6 noART: 8	NR	ART: 0 noART: 6	ART: 10 noART: 0
Kumar, 2013 [6]	India	326	0	1–14 yr	100	NR	61.7	PC+EC 20.9, AC 16.6	8.3	7.7	5.5	2.8	NR	NR	NR
Nabbanja, 2013 [18]	Uganda	368	0	1.5–17 yr	67,4	35% 0–2 yr 40% 3–5 yr 25% 6–9 yr	77.4	PC 50.5, EC 10.3	4.1	0.3	NR	4.1	0.5	4.3	NR
Ribeiro, 2013 [8]	Brazil	57	0	3 mth–14 yr	NR	NR	69.6	PC 3.5, EC 3.5, AC 8.8	57.9	NR	7	7	NR	NR	NR
Subramaniam, 2013 [4]	India	150	0	6–18 yr	55.3	NR	99.3	PC 68.7, EC 22, AC 96.6	NR	NR	6.7	33.3	NR	0.7	14.7
Meless, 2014 [19]	West Africa	420	0	5–15 yr	100	4.5 yr	8.3	4.8	1.7		NR	1	1.2	NR	NR
Oliscovicz, 2015 [20]	Brazil	111	0	2–16 yr	87.4	35 mth	23.4	1.8	15.3	NR	5.4	1.8	0.9	NR	NR
Oyedeji, 2015 [3]	Nigeria	58	0	3 mth–13 yr	63.8	NR	63.8	17.2	3.4	NR	5.2	NR	6.8	NR	NR
Subramaniam, 2015 [21]	India	221	0	6–18 yr	49.3	NR	NR	PC 63.8, EC 22.6, AC 81.9	NR	NR	9	30.3	NR	0.9	15.4
Divakar, 2015 [22]	India	62	55	5–15 yr	52.99	>3 yr	ART: 19.6 noART: 30.7	ART: 2.8 noART: 38.2	NR	NR	ART:0 noART: 5.5	ART: 5.7 noART: 7.3		ART: 0 noART: 5.5	ART: 11.4 noART: 14.5

Legend. n: number of HIV-positive children; ctr: control population; OL: oral lesions; OC: oral candidiasis (EC: erythematous candidiasis, PC: pseudomembranous candidiasis, AC: angular cheilitis, HC: hyperplastic candidiasis); G: gingivitis; P: periodontitis; LGE: linear gingival erythema; ULC: ulcers; HSV: herpes lesions; OHL: oral hairy leucoplakia; PIG: pigmentation; and NR: not reported.

**Table 3 pathogens-09-00088-t003:** DMFT and dmft indices in HIV-positive children.

Author, Year	*n*	DMFT	dmft
Sales-Peres, 2010	90	0.6	2.6
Nabbanja, 2013	368	2.7	11.8
Ribeiro, 2013	57	3.41	7.01
Meless, 2014	420	4.9	NR
Oliscovicz, 2015	111	1.9	3.1

n: number of HIV-positive children.

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
