# Peer review of "Oral Manifestations in HIV-Positive Children: A Systematic Review"

_pathogens, 2020, doi:10.3390/pathogens9020088_

Round 1
Author Response
2020, 22h January
Dear Reviewer,
Thank you for your kind comments and suggestions.
We appreciate your time in reviewing our paper.
Yours sincerely,
Dorina Lauritano

Reviewer 2 Report
37 references is not enough for a systematic review.
I thank it is ineffective to select only PubMed and Scopus like database.
Author Response
2020, 22h January
Dear Reviewer,
Thank you for your kind comments and suggestions. We have increased the number of references and extended our research to other databases such as Web of Science, Embase, Google scholar, Sciforum, Encyclopedia, Susy, etc.
We appreciate your time in reviewing our paper.
Yours sincerely,
Dorina Lauritano

Reviewer 3 Report
Thank you for this valuable and interesting work that assessed the literature for the oral manifestations in children with HIV. I have only few comments which I believe will improve the quality of your research:
The manuscript would benefit from proofreading throughout all the sections. In the Results section, a number of studies has been excluded ''because the full text was not available''. It would have been beneficial if these studies were found with effort (through some national libraries) to avoid missing some data. It is recommended to add a table in the appendix with the excluded articles and the reasons for exclusion. Has this systematic review been reported according to PRISMA checklist? Has any quality assessment for eligible studies been conducted in this systematic review and reported in a table? Some sections in the Results had justifications whilst this should be in the Discussion.Author Response
2020, 22th January
Dear Reviewer,
Thank you for your kind comments and suggestions. We have modified our review as follows:
In the Results section, a number of studies has been excluded ''because the full text was not available''. It would have been beneficial if these studies were found with effort (through some national libraries) to avoid missing some data. It is recommended to add a table in the appendix with the excluded articles and the reasons for exclusion.The electronic research of this review began in November 2018 and ended in March 2019. Following your suggestion, we conducted new research, in order to select those studies, which were excluded from the study. However, the electronic research to date does not provide the same results and for this reason, it was impossible to add the table with the excluded articles.
Has this systematic review been reported according to the PRISMA checklist?
We have modified our paper following the PRISMA statement: “To provide an appropriate search protocol, methods and inclusion criteria of this systematic review were selected following the PRISMA statement [9]”.
[9]. Liberati A, Altman DG, Tetzlaff J, Mulrow C, Gøtzsche PC, Ioannidis JP, Clarke M, Devereaux PJ, Kleijnen J, Moher D. (2009). The PRISMA statement for reporting systematic reviews and meta-analyses of studies that evaluate health care interventions: explanation and elaboration. J Clin Epidemiol, 62(10):e1-34; doi: 10.1016/j.jclinepi.2009.06.006.
Has any quality assessment for eligible studies been conducted in this systematic review and reported in a table?
We have performed the quality assessment using the Newcastle-Ottawa scale: Quality assessment of the selected studies was conducted using the Newcastle-Ottawa scale (NOS) [11]. The highest and the lowest score recorded were equal to 8 and 5 respectively (average score 6.73). Almost all the samples selected by the included articles were truly or somewhat representative of the average in the target population and all the items assessed the oral health of each subject using validated measurement tools. Data were analyzed with appropriate and well described statistical tests (including confidence intervals and the probability level) by all the authors. So, the quality of the studies may be considered high.
[11] GA Wells, B Shea, D O’Connell, J Petersen, V Welch, M Losos, P Tugwell. The Newcastle-Ottawa Scale (NOS) for assessing the quality of nonrandomized studies in meta-analyses. Dep. Epidemiol. Community Med. Univ.
Table 1. Quality assessment of included studies
Studies |
Representativeness of the sample |
Sample size |
Non-respondents |
Ascertainment of the exposure |
Comparability |
Outcome |
Total |
Ranganathan, 2010 |
+ |
- |
- |
+ |
+- |
++ |
5 |
Sales-Peres, 2010 |
+ |
+ |
+ |
+ |
+- |
++ |
7 |
Rwenyonyi, 2011 |
+ |
+ |
+ |
+ |
+- |
++ |
7 |
Adebola, 2012 |
+ |
+ |
- |
+ |
+- |
++ |
6 |
Ponnam, 2012 |
+ |
+ |
+ |
+ |
++ |
++ |
8 |
Baghirath, 2013 |
+ |
+ |
- |
+ |
++ |
++ |
7 |
Kumar, 2013 |
+ |
+ |
- |
+ |
++ |
+- |
6 |
Nabbanja, 2013 |
+ |
+ |
- |
+ |
++ |
+- |
6 |
Ribeiro, 2013 |
+ |
- |
+ |
+ |
++ |
++ |
7 |
Subramaniam, 2013 |
+ |
+ |
+ |
+ |
+- |
++ |
7 |
Meless, 2014 |
+ |
+ |
+ |
+ |
+- |
++ |
7 |
Oliscovicz, 2015 |
+ |
+ |
+ |
+ |
+- |
++ |
7 |
Oyedeji, 2015 |
+ |
+ |
- |
+ |
+- |
++ |
6 |
Subramaniam, 2015 |
+ |
+ |
+ |
+ |
+- |
++ |
7 |
Divakar, 2015 |
+ |
+ |
+ |
+ |
++ |
++ |
8 |
+ = star assigned; - = star not assigned
Some sections in the Results had justifications whilst this should be in the Discussion.
We reorganized the references, removing them from the Results and Conclusion section and adding them in the Discussion one.
We appreciate your time and look forward to your response.
Yours sincerely,
Dorina Lauritano

Reviewer 4 Report
The authors present a systematic review about Oral manifestations in hiv-positive children: a systematic review. The article is well written however there are important and major revisions to perform. Indeed the search strategy and the study selection should be improved.
Specifically:
- Page 2 line 71: since this is a systematic review, please you have to perform the search also in EMBASE and Cochrane Central databases and not only in Pubmed and Scopus. This type of “systematic search” is needed to perform a correct “systematic review”
- In material and methods, please include the keywords that you used to perform the search in Pubmed, Scopus, EMBASE and Cochrane Central databases.
- Did you follow the Meta-analysis of Observational Studies in Epidemiology (MOOSE) proposal and the Preferred Reporting Items for Systematic Reviews and Meta-Analyses (PRISMA) guidelines, where feasible? Please specify.
- Did you contact the experts and perused reference sections in order to identify all relevant reports as well as unpublished data? In case of true systematic review this is needed.
- In Materials and Methods, please specify the last and exact day in March 2019 for the search
- Why you decided to evaluate only 10 years? (…”published in English from 2010 to date..”). If this is a systematic review you have to include all the possible articles present in the databases EMBASE, Cochrane Central databses, Pubmed and Scopus
- In page 2 line 84 you wrote “…We only considered the studies following these inclusion criteria: sample of patients under 18”, but in page 2 line 89 you wrote “…we have decided to include older children up to 18 years of age”. This is contradictory. Please specify
- In page 2 line 89 you wrote …”In most countries”, which ones? Please, specify and add the reference
- Please change the flowchart (Figure 1), adding also the articles present in EMBASE and Cochrane Central databases
I, conclusion I have doubt about this systematic review, since as presented cannot be defined as a systematic review. Please revise the article (above all the search strategy) and I will be happy to review the revised article. Thank you.
Author Response
2020, 22h January
Dear Reviewer,
Thank you for your kind comments and suggestions. Following your indications we have modified our paper as follows:
Page 2 line 71: since this is a systematic review, please have to perform the search also in EMBASE and Cochrane Central databases and not only in Pubmed and Scopus. This type of “systematic search” is needed to perform a correct “systematic review”.Please change the flowchart (Figure 1), adding also the articles present in EMBASE and Cochrane Central databases. Why did you decide to evaluate only 10 years? (…” published in English from 2010 to date..”). If this is a systematic review you have to include all the possible articles present in the databases EMBASE, Cochrane Central database, Pubmed and Scopus.
We have performed the search also in EMBASE and Cochrane Central, considering all the possible present articles present in these databases and we add one article to our review: Divakar D.D., Al Kheraif A.A., Ramakrishnaiah R., Khan A.A., Sandeepa N.C., Alshahrani O.A., Alahmari A. (2015). Oral manifestations in human immunodeficiency virus-infected pediatric patients receiving and not receiving antiretroviral therapy: A cross-sectional study. Paediatria Croatica 2015 59:3 (152-158).
We have also modified the flowchart, including the papers found in EMBASE and Cochrane Central.
In material and methods, please include the keywords that you used to perform the search in Pubmed, Scopus, EMBASE and Cochrane Central databases.
We included the keywords used to perform the search in the section Material and Methods: “The following search items, combined with the Boolean term “AND”, were used to perform an electronic search in the PubMed, Cochrane Central Library, EMBASE, and Scopus databases: oral lesions, oral manifestations, HIV infection (Mesh database terms) and HIV-positive children, HIV-positive pediatric patients”
Did you follow the Meta-analysis of Observational Studies in Epidemiology (MOOSE) proposal and the Preferred Reporting Items for Systematic Reviews and Meta-Analyses (PRISMA) guidelines, where feasible? Please specify.
We reorganized our paper, following the PRISMA statement: “To provide an appropriate search protocol, methods and inclusion criteria of this systematic review were selected following the PRISMA statement [9]”.
[9] Liberati A, Altman DG, Tetzlaff J, Mulrow C, Gøtzsche PC, Ioannidis JP, Clarke M, Devereaux PJ, Kleijnen J, Moher D. (2009). The PRISMA statement for reporting systematic reviews and meta-analyses of studies that evaluate health care interventions: explanation and elaboration. J Clin Epidemiol, 62(10):e1-34; doi: 10.1016/j.jclinepi.2009.06.006.
Did you contact the experts and perused reference sections in order to identify all relevant reports as well as unpublished data? In the case of a true systematic review, this is needed.
We contacted experts and we analyzed the reference sections of each included paper: “Experts were employed in order to identify any unpublished data and reference sections of all the selected studies were analyzed to avoid the exclusion of relevant papers”.
In Materials and Methods, please specify the last and exact day in March 2019 for the search.
We have specified the last exact day in March 2019 for the search: “The search began in November 2018 and ended on 31 March 2019”
In page 2 line 84 you wrote “…We only considered the studies following these inclusion criteria: sample of patients under 18”, but in page 2 line 89 you wrote, “…we have decided to include older children up to 18 years of age”. This is contradictory. Please specify.
We have modified the first point of the Inclusion Criteria as follows: “Sample of HIV positive patients from the age of 0 to 18 (with wide age range and without significant gender predominance)”.
In page 2 line 89 you wrote …”In most countries”, which ones? Please, specify and add the reference.
We specified the countries in which young people less than 18 years are considered minor: “In most countries, such as Europe, Australia, Mexico, Canada, and Russia, in fact, young people less than 18 years are considered minor and, even if modes of infection are different in adolescents than in younger children or infants, in the majority of the studies on this topic also adolescents are included”
We appreciate your time and look forward to your response.
Yours sincerely,
Dorina Lauritano

Round 2
Reviewer 2 Report
Good work in a short time
Reviewer 4 Report
The article can be now accepted for publication